# YAP/TAZ Promote GLUT1 Expression and Are Associated with Prognosis in Endometrial Cancer

**DOI:** 10.3390/cancers17152554

**Published:** 2025-08-01

**Authors:** Masayuki Fujita, Makoto Orisaka, Tetsuya Mizutani, Yuko Fujita, Toshimichi Onuma, Hideaki Tsuyoshi, Yoshio Yoshida

**Affiliations:** 1Department of Obstetrics and Gynecology, University of Fukui, Fukui 910-1193, Japan; fjt@u-fukui.ac.jp (M.F.); orisaka@u-fukui.ac.jp (M.O.); yuchifu@u-fukui.ac.jp (Y.F.); toonuma@u-fukui.ac.jp (T.O.); gth@u-fukui.ac.jp (H.T.); 2Department of Nursing, Faculty of Nursing and Welfare Sciences, Fukui Prefectural University, Fukui 910-1195, Japan; mizutani@fpu.ac.jp

**Keywords:** endometrial cancer, GLUT1, hippo pathway, TAZ, YAP

## Abstract

The Hippo pathway is a key cancer-related signaling cascade that regulates cell proliferation. Yes-associated protein (YAP) and transcriptional coactivator with PDZ-binding motif (TAZ) act as downstream effectors within this pathway. Glucose transporter (GLUT) proteins also contribute to cellular proliferation. This study investigated the link between YAP/TAZ and GLUT1 in endometrial carcinoma. In cancer cell lines, YAP/TAZ were identified as key regulators of GLUT1 expression. An analysis of 100 patient tissue samples revealed a significant correlation between TAZ and GLUT1 expression. Moreover, nuclear TAZ expression was associated with poor progression-free survival (*p* < 0.05). The study concludes that YAP/TAZ promote tumor growth via GLUT1, suggesting that targeting these proteins could be a promising therapeutic strategy for future treatments.

## 1. Introduction

Endometrial cancer is the sixth most common malignant tumor in women [1], and obesity and long-term estrogen supplementation are considered risk factors. The incidence of endometrial cancer is increasing worldwide as obesity rates increase [2]. Although the 5-year survival rate for endometrial cancer is good, at 81%, the prognosis of recurrent and advanced cancer is extremely poor [2]. Therefore, it is important to identify new proteins and signaling pathways associated with increased malignancy and develop new therapeutic drugs.

Recent research has revealed that glucose transporter (GLUT) proteins and Yes-associated protein (YAP)/transcriptional coactivator with PDZ-binding motif (TAZ) are associated with malignant tumors. In tumor cells, the Warburg effect—an established alteration in glucose metabolism—enhances glycolysis and glucose uptake for ATP production, even under aerobic conditions. This metabolic shift leads to increased expression of GLUTs, which has been associated with tumor proliferation and aggressiveness [3]. For example, GLUT1 (encoded by *solute carrier family 2 member 1* (*SLC2A1*)) is overexpressed in many malignant tumors, including uterine cancer, ovarian cancer, cervical cancer, and lung cancer [3], and its expression is reportedly associated with prognosis [4].

YAP/TAZ are transcriptional coactivators at the core of the Hippo signaling pathway. The Hippo pathway is a signaling pathway that regulates cell proliferation, differentiation, and survival and plays an important role in organ development and homeostasis. The Hippo pathway is composed of several key components, including mammalian STE20-like kinase 1/2 (MST1/2), protein Salvador homologue 1 (SAV1), MOB kinase activator 1A/B (MOB1A/B), large tumor suppressor kinase 1/2 (LATS1/2), YAP, TAZ, and the transcriptional enhanced associated domain (TEAD) family 1 [5]. In particular, YAP/TAZ exert their transcriptional activity primarily by binding to TEAD and promoting the expression of target genes. Recent research has revealed that abnormalities in the Hippo pathway, specifically the overexpression of YAP/TAZ, cause excessive proliferation of malignant tumors [6]. In addition, overexpression of YAP/TAZ has been observed in endometrial cancer and associated with prognosis [7].

Both GLUT and YAP/TAZ are involved in the proliferation of malignant tumors. For example, YAP reportedly enhances GLUT3 expression in metastatic colorectal cancer and promotes metastasis [8]. YAP also promotes GLUT1 expression and glucose uptake in hepatocyte-specific Yap transgenic model mice and breast cancer cells [9,10], although similar effects in endometrial cancer have not been reported.

In this study, we investigated the effect of YAP/TAZ on GLUT1 expression using two cell lines and clinical specimens derived from cases of endometrial cancer. As our results suggest that YAP/TAZ would be useful therapeutic targets in endometrial cancer, we investigated the clinical relevance of YAP/TAZ in endometrial cancer and their relationship to prognosis.

## 2. Materials and Methods

### 2.1. Cell Culture

HHUA and Ishikawa cells, which are human endometrial cancer cell lines, were obtained from the RIKEN BioResource Center (Tsukuba, Japan) and the JCRB Cell Bank (Osaka, Japan), respectively. HHUA cells were maintained in DMEM/F-12 medium (FUJIFILM Wako Pure Chemical Co., Osaka, Japan) supplemented with 15% fetal bovine serum and 1% penicillin–streptomycin (FUJIFILM Wako Pure Chemical Co.). Ishikawa cells were maintained using RPMI1640 (FUJIFILM Wako Pure Chemical Co.) supplemented with 10% fetal bovine serum (FBS) and 1% penicillin–streptomycin. Cells were cultured at 37 °C in a humidified atmosphere with 5% CO_2_ and passaged twice weekly.

### 2.2. Small Interfering RNA (siRNA) Knockdown and Adenovirus Infection

One day before transfection, cells were seeded in 6-well plates (2 mL) at 1.4 × 105 cells/mL for HHUA cells or 0.7 × 105 cells/mL for Ishikawa cells. The cells were transfected with target-specific or non-targeting siRNAs using Lipofectamine RNAiMAX (Thermo Fisher Scientific, Waltham, MA, USA) and further incubated for 48 h. The final siRNA concentration in the medium was 20 nM. The sequences of siRNAs used in the study are listed in Appendix A [11,12,13,14]. The study also employed a constitutively active mutant form of YAP that contains 5 serine-to-alanine (5SA) mutations that abolish phosphorylation sites in wild-type YAP, known as YAP-5SA [15,16]. An adenovirus that expresses YAP-5SA was described previously [17]. An adenovirus that expresses green fluorescent protein was used as a control. Cells were infected with adenovirus at the indicated multiplicity of infection (MOI), which equals the number of viral particles per cell. To combine siRNA knockdown and adenovirus-mediated transduction experiments, adenovirus infection was performed 4 h after siRNA transfection, followed by additional incubation for 48 h [16,17].

### 2.3. RNA Isolation and RT-qPCR Analysis

Total RNA was isolated from cells using Isogen II according to the manufacturer’s instructions (Nippon Gene, Tokyo, Japan). Three micrograms of total RNA per 60 µL was reverse-transcribed using ReverTra Ace qPCR RT Master Mix with gDNA Remover (TOYOBO, Osaka, Japan), and a portion (25 ng equivalent of total RNA) of the reaction mixture was subjected to qPCR analysis. Quantitative PCR was performed using Thunderbird SYBR qPCR Mix (TOYOBO) in duplicate, and the data were analyzed using a StepOnePlus™ Real-Time PCR System (Thermo Fisher Scientific) by standard curve quantification, as described previously [16]. The sequences of primers used in the study are listed in Appendix A.

### 2.4. Patients

This retrospective study included 100 patients diagnosed with uterine endometrial cancer at the University of Fukui Hospital, between 2017 and 2023. Formalin-fixed, paraffin-embedded tissue specimens were obtained from all patients and analyzed retrospectively. Information regarding clinical and pathological factors was extracted from medical and pathology records. Patients histologically diagnosed with uterine endometrial cancer were included in the study, and definitive histopathological diagnoses were made by two certified pathologists. Patients were treated according to the clinical guidelines of the Japanese Society of Gynecologic Oncology and followed up until death or for at least 9 months from the date of first consultation. The study was approved by the Institutional Review Board (IRB) of the University of Fukui Hospital (approval number: 20230163).

### 2.5. Tissue Samples and Immunohistochemistry

Formalin-fixed, paraffin-embedded tissues were immunohistochemically stained using Nichirei’s Histofine Series and Simple Stain MAXPO (MULTI) (Nichirei Biosciences Inc., Tokyo, Japan). Sections (2.5-μm thickness) were deparaffinized three times for 15 min each using ClearPLUS, rehydrated via alcohol series, and subjected to antigen retrieval for 15 min in 10 mM citrate buffer (pH 6.0). After cooling, the sections were washed three times with phosphate-buffered saline (PBS, pH 7.2). Endogenous peroxidase activity was blocked by immersion in 3% hydrogen peroxide for 20 min. Nonspecific binding of the primary antibody was blocked by incubating sections with diluted normal serum (Blocking one Histo (Nakarai Tesque Inc., Kyoto, Japan)) for 10 min at room temperature. The sections were then incubated overnight or for 1 h (GLUT1) with primary antibodies against GLUT1 (rabbit polyclonal antibody (1:200; Abcam, Cambridge, UK) and TAZ (rabbit polyclonal antibody (1:200; Sigma-Aldrich St. Louis, MO, USA). Sections were incubated in the same manner with primary antibodies against YAP (rabbit monoclonal antibody (1:200; Cell Signaling Technology, Danvers, MA, USA). Primary antibodies were diluted in PBS containing 1% bovine serum albumin. After washing with PBS, Simple Stain MAXPO (MULTI) (Nichirei Biosciences Inc., Tokyo, Japan) was added as a secondary antibody and incubated for 30 min. Signals were visualized using a chromogen solution containing substrate and Betazoid DAB Chromogen kit (Biocare Medical, Concord, CA, USA). Sections were counterstained with Mayer’s acid hematoxylin and washed multiple times with 70–100% alcohol. After treatment with Lemosol (FUJIFILM Wako Pure Chemical Corporation, Osaka, Japan) the sections were covered.

To determine staining intensity, sections of corpora lutea were stained as positive controls for YAP and TAZ, and the cell membrane of red blood cells was stained as a positive control for GLUT1 (Appendix A) [18,19]. The intensity and distribution of immunohistochemical staining of YAP, TAZ, and GLUT1 were evaluated using a semiquantitative method (immunoreactivity score (IRS)) [20], which was calculated as follows: IRS = signal intensity (SI) × percentage of positive cells (PP) (SI graded as 0 = no staining, 1 = weak staining, 2 = moderate staining, and 3 = strong staining, and PP was scored based on the calculated immunostaining-positive area of the tumor section as 0 = no staining, 1 = less than 10% staining, 2 = 11–50% staining, 3 = 51–80% staining, and 4 = ≥81% staining). Immunohistochemical staining was scored by two independent observers. Representative images of immunostaining analyses are shown in Figure 1.

### 2.6. Statistical Analysis

RT-qPCR results for transfected and control HHUA and Ishikawa cells were analyzed using the Sidak multiple-comparisons method. Outcome measures for clinical samples included progression-free survival (PFS) and overall survival (OS). Both PFS and OS were assessed from the date of first consultation. Tumor recurrence was confirmed by imaging. Sample size calculations were based on previous results regarding estrogen receptor and progesterone receptor expression in endometrial cancer as predictive markers for PFS [21]. To evaluate the correlation between nuclear YAP, nuclear TAZ, and GLUT1 expression, the IRSs were analyzed using the Pearson correlation coefficient. The relationship between IRS of each protein and clinical characteristics was analyzed using the Mann–Whitney *U* test. The area under the receiver operating characteristic (ROC) curve (AUC) was calculated for each protein to determine the optimal cutoff value for high-accuracy discrimination. The relationship between each protein and PFS and OS was evaluated using Kaplan–Meier curves, and statistical significance was calculated using the log-rank test. Cox proportional-hazards regression models were used for univariate and multivariate analyses. Significance was defined as *p* < 0.05 (two-tailed test). Statistical analyses were performed using EZR version 1.68 (Jichi Medical University Saitama Medical Center, Saitama, Japan) [22] for ROC curve analysis and GraphPad Prism 10 for others.

## 3. Results

### 3.1. YAP and TAZ Positively Regulate GLUT1 Expression in Endometrial Cancer Cells

In this study, we analyzed the effect of YAP and TAZ on GLUT1 expression and the underlying molecular mechanism using HHUA and Ishikawa cells derived from endometrial cancers. siRNAs were used to knockdown the expression of YAP, TAZ, or both YAP and TAZ (YAP/TAZ) in HHUA and Ishikawa cells, and the resulting changes in GLUT1 expression were monitored using RT-qPCR. Knockdown of YAP, TAZ, or YAP/TAZ significantly decreased the expression of both GLUT1 and cysteine-rich angiogenic inducer 61 (CYR61), a typical target gene of YAP/TAZ–TEAD transcription factors (*p* < 0.0001, Figure 2). To confirm that GLUT1 expression is regulated by YAP/TAZ, we investigated the effects of forced YAP expression. Since YAP and TAZ are negatively regulated by phosphorylation, we used a phosphorylation-deficient mutant of YAP (YAP-5SA), which functions as a constitutively active form of YAP. In addition, to provide evidence that both YAP and TAZ play critical roles in GLUT1 expression, combined transduction of YAP-5SA and TAZ knockdown experiments were conducted. In cells transfected with negative control (HHUA cells) or TAZ-specific siRNA (both HHUA and Ishikawa cells), GLUT1 expression was significantly induced by the introduction of YAP-5SA, similar to CYR61 (*p* < 0.05, Figure 3). The rate of GLUT1 induction by YAP-5SA was greater in TAZ-knockdown cells than negative control cells, suggesting that YAP-5SA expression compensates for the loss of TAZ function. These results suggest that YAP and TAZ are important factors that positively regulate GLUT1 expression in endometrial cancer cells and that these pathways may be involved in the regulation of cancer cell metabolism.

### 3.2. Patient Characteristics

Table 1 summarizes clinical information for the 100 patients included in this study. The median age at diagnosis was 61.2 years (range, 35–89 years). With regard to pathology, the histologic subtypes were endometrioid carcinoma (n = 90), mixed adenocarcinoma (n = 1), serous adenocarcinoma (n = 4), carcinosarcoma (n = 1), clear-cell carcinoma (n = 2), mesonephric adenocarcinoma (n = 1), and small cell carcinoma (n = 1). Ninety-nine patients (99%) underwent simple hysterectomy, modified radical hysterectomy, or radical hysterectomy with bilateral salpingo-oophorectomy, and 50 patients (50%) underwent pelvic or para-aortic lymphadenectomy. One patient (1%) received progesterone therapy. Thirty-eight patients (38%) (stage IB or higher, stage IA with endometrioid G3 or other histology, or positive for vascular invasion) underwent adjuvant chemotherapy according to Chapter 3 of the 2018 Clinical Guidelines of the Japan Society of Gynecological Oncology [23]. The median follow-up period was 36 months (range, 9–76 months). During the follow-up period, 11 patients (11%) experienced recurrence, and 5 patients (5%) died. To investigate the correlation between nuclear and cytoplasmic expression of YAP and TAZ and the expression of GLUT1 with respect to the clinical data and the resulting impact on prognosis, IRSs were calculated based on the results of immunohistochemical staining.

### 3.3. Correlation of YAP, TAZ, and GLUT1 Expression and Association of Each Protein with Clinical Parameters

The mean IRS for nuclear YAP, cytoplasmic YAP, nuclear TAZ, cytoplasmic TAZ, and GLUT1 was 4.24 ± 2.599 (0–12), 6.52 ± 3.108 (0–12), 4.32 ± 2.703 (0–12), 5.47 ± 2.804 (0–12), and 3.80 ± 2.125 (0–12). These IRSs were analyzed using the Pearson correlation coefficient to determine the correlation between nuclear YAP, nuclear TAZ, and GLUT1 expression. For nuclear YAP and GLUT1 expression and nuclear TAZ and GLUT1 expression, the results were as follows: y = −0.05291x + 4.438, r = −0.04267, *p* = 0.6734 and y = 0.3295x + 3.069, r = 0.2591, *p* = 0.0093, respectively (Figure 4). These results suggest that TAZ and GLUT1 expression are positively correlated in endometrial cancer clinical specimens.

We also examined the relationships between the IRS for each protein and various clinical parameters. In addition, the difference in IRS (nuclear expression minus cytoplasmic expression: N-C) was determined to compare the nuclear and cytoplasmic expression of YAP and TAZ. High nuclear YAP expression tended to be associated with <1/2 myometrial invasion (*p* = 0.0934) and significantly correlated with endometrioid cancer G1 and G2 (*p* = 0.0201) and negative vascular invasion (*p* = 0.0448). High cytoplasmic YAP expression tended to be associated with negative vascular invasion (*p* = 0.0754) and was significantly correlated with FIGO stage I–II (*p* = 0.0197), <1/2 myometrial invasion (*p* = 0.0179), tumor diameter < 2 cm (*p* = 0.0238), and negative lymph-node metastasis (*p* = 0.0493). Higher nuclear YAP expression than cytoplasmic YAP expression tended to be associated with tumor diameter of ≥2 cm (*p* = 0.075). High nuclear TAZ expression also tended to be associated with age ≥ 50 years (*p* = 0.0825). High cytoplasmic TAZ expression was significantly associated with age ≥ 50 years (*p* = 0.009). Higher nuclear TAZ expression than cytoplasmic TAZ expression tended to be associated with FIGO stage III–IV (*p* = 0.0537), endometrioid carcinoma G3 or other histological types (*p* = 0.0951), and muscle layer invasion ≥ 1/2 (*p* = 0.0833), and it was significantly associated with positive lymph-node metastasis (*p* = 0.0218). High GLUT1 expression tended to be associated with a tumor diameter of ≥2 cm (*p* = 0.094) and was significantly correlated with age ≥ 50 years (*p* = 0.0248) and positive vascular invasion (*p* = 0.0421) (Table 2 and Table 3). These results demonstrate that high YAP and TAZ nuclear and cytoplasmic expression are not correlated with clinical parameters generally considered to indicate a poor prognosis and instead suggest that TAZ expression may be a biomarker of poor prognosis when localized in the nucleus.

### 3.4. Correlation Between YAP and TAZ Expression and Clinicopathologic Parameters

In this study, we evaluated the effects of YAP and TAZ expression on PFS and OS and identified cutoff values useful for prognostic prediction using ROC analysis. The cutoff value for PFS of nuclear YAP was 3.5 (AUC, 0.644, 72.7% sensitivity, 60.7% specificity). The cutoff value for PFS of cytoplasmic YAP was 5 (AUC, 0.594, 54.5% sensitivity, 67.4% specificity). The cutoff value for PFS of nuclear TAZ was 0.5 (AUC, 0.563, 27.3% sensitivity, 98.9% specificity). The cutoff value for PFS of cytoplasmic TAZ was 1.5 (AUC, 0.635, 27.3% sensitivity, 95.5% specificity). The cutoff value for PFS of YAP N-C was 0 (AUC, 0.502, 36.4% sensitivity, 83.1% specificity). The cutoff value for PFS of TAZ N-C was −0.5 (AUC, 0.692, 72.7% sensitivity, 66.3% specificity) (Figure 5). The cutoff value for OS of nuclear YAP was 3.5 (AUC, 0.629, 80.0% sensitivity, 58.9% specificity). The cutoff value for OS of cytoplasmic YAP was 4 (AUC, 0.585, 60.0% sensitivity, 72.6% specificity). The cutoff value for OS of nuclear TAZ was 1.5 (AUC, 0.675, 60.0% sensitivity, 87.4% specificity). The cutoff value for OS of cytoplasmic TAZ was 6 (AUC, 0.751, 100% sensitivity, 38.9% specificity). The cutoff value for OS of YAP N-C was −3 (AUC, 0.523, 60.0% sensitivity, 62.1% specificity). The cutoff value for OS of TAZ N-C was 0 (AUC, 0.705, 80.0% sensitivity, 72.6% specificity) (Figure 6). TAZ N-C had the largest AUC for PFS (*p* = 0.039). Cytoplasmic TAZ had the largest AUC for OS (*p* = 0.060).

### 3.5. Correlation Between YAP and TAZ Expression and Prognosis

Kaplan–Meier analyses showed that patients with high YAP nuclear expression, high TAZ nuclear expression, and high TAZ cytoplasmic expression had significantly longer PFS (*p* < 0.05). Patients with low TAZ nuclear expression also had significantly shorter OS (*p* < 0.001). By contrast, patients with a high TAZ N-C (nuclear expression > cytoplasmic expression) had significantly shorter PFS (*p* = 0.01) (Figure 7 and Figure 8).

In the univariate analysis, high nuclear YAP expression was associated with longer PFS (*p* = 0.042), and high TAZ N-C (nuclear expression > cytoplasmic expression) was associated with shorter PFS (*p* = 0.0193). In the multivariate analysis, FIGO stage III–IV, endometrioid G3, and no endometrioid adenocarcinoma histology were identified as independent prognostic factors for PFS (*p* < 0.05) (Table 4). FIGO stage III–IV was the only independent prognostic factor for OS (*p* = 0.0021) (Table 5).

## 4. Discussion

Using two cell lines derived from endometrial cancer, the present study demonstrated that YAP/TAZ play an important role in the regulation of GLUT1 expression. Analyses of tissues from endometrial cancer patients revealed a significant correlation (*p* < 0.05) between TAZ and GLUT1 expression, and biased nuclear expression of TAZ was associated with poor prognosis. These results suggest that YAP/TAZ promote tumor growth via GLUT1, and it is particularly noteworthy that biased nuclear expression of TAZ is strongly associated with prognosis. Furthermore, these results suggest the potential usefulness of therapeutic strategies targeting YAP/TAZ; thus, the development of inhibitors and the implementation of clinical trials are expected.

YAP and TAZ are important regulators of the Hippo signaling pathway and strongly associated with the progression of malignant tumors. YAP and TAZ have very similar structures and interact with TEAD1-4 and other transcription factors. Thus, YAP and TAZ share the same mechanism for regulating gene expression [24]. The Hippo pathway is one of eight known cancer-signaling pathways identified through analyses of >9000 cancers [25]. Abnormalities in the Hippo pathway (e.g., overexpression of YAP/TAZ) have been reported in various malignant tumors [6]. Overexpression of YAP/TAZ promotes the progression of malignant tumors by inducing cancer cell proliferation and suppressing apoptosis [26,27]. Overexpression of YAP/TAZ is also reportedly associated with prognosis in breast cancer and hepatoblastoma [28,29]. In estrogen-dependent endometrioid carcinoma, YAP expression was significantly associated with malignancy grade, stage, lymphatic invasion, and OS [7]. Thus, YAP/TAZ may be involved in the progression of uterine endometrial cancer.

In malignant tumors, ATP is produced via glycolysis through mitochondrial oxidative phosphorylation in not only anaerobic but also aerobic environments (Warburg effect) [30]. Therefore, GLUT1 expression is upregulated in many malignant tumors [3]. In a study of endometrial cancer, Goldman et al. investigated the expression of GLUT1 in normal endometrial tissue and 65 endometrial cancer tissue specimens from hysterectomized uteruses and reported that GLUT1 expression increased as the tissue became less differentiated histologically; expression of GLUT1 in serous adenocarcinoma was twice that in well-differentiated adenocarcinoma (G1) (*p* = 0.002) [31]. In other words, GLUT1 may also be involved in the progression of endometrial cancer.

Regarding the correlation between the expression of YAP/TAZ and GLUT, recent reports have shown that YAP enhances GLUT3 expression in metastatic colorectal cancer, thereby promoting metastasis [8], and that YAP promotes GLUT1 expression and glucose uptake in hepatocyte-specific Yap transgenic model mice and breast cancer cells [9,10]. Therefore, YAP/TAZ may also be associated with glucose transporters in endometrial cancer.

Based on the abovementioned results, we therefore employed HHUA and Ishikawa cells to examine the effects on GLUT1 gene expression of siRNA-mediated knockdown of YAP, TAZ, or YAP/TAZ and adenovirus-mediated overexpression of a constitutively active mutant form of YAP. In addition, tissue samples taken from endometrial cancer patients were analyzed using immunohistochemistry to examine the relationship between nuclear and cytoplasmic expression of YAP and TAZ and the resulting expression of GLUT1, as well as the relationship with various clinical parameters (patient age, stage of disease, histological type, etc.) and prognosis.

In endometrial cancer cells, as in other cells, CYR61 expression was YAP/TAZ dependent. By contrast, both YAP and TAZ were shown to positively regulate GLUT1 expression (Figure 2 and Figure 3). The effect of YAP/TAZ on GLUT1 expression was smaller than that of CYR61 in loss- and gain-of-function studies, but this may have been because GLUT1 is ubiquitously expressed and less dependent on YAP/TAZ than CYR61. In addition, a significant correlation between TAZ and GLUT1 expression was observed in the immunohistochemical analysis of clinical specimens (*p* < 0.01), suggesting that at least TAZ contributes to the progression of endometrial cancer by inducing GLUT1 expression. Although GLUT1 expression is regulated by multiple oncogenic pathways, including Hypoxia-inducible factor 1 alpha and KRAS signaling [32,33], our results suggest a direct regulatory link between the Hippo pathway and GLUT1 expression. Previous studies have shown that YAP/TAZ can transcriptionally activate GLUT1 through TEAD-binding motifs in the promoter region [9]. In our study, YAP/TAZ knockdown led to a marked reduction in GLUT1 protein level in endometrial cancer cells, supporting this regulatory relationship. Nevertheless, we acknowledge that other pathways may also contribute to GLUT1 regulation in the tumor microenvironment, and further studies using in vivo models or patient-derived samples are warranted to confirm the specific role of YAP/TAZ in this context.

With regard to patient outcomes, PFS was significantly shorter when TAZ expression was biased toward the nucleus (*p* = 0.01). YAP/TAZ activity in the Hippo pathway is controlled by multiple kinases, which can lead to changes in intracellular localization [34]. When the Hippo pathway is activated, YAP/TAZ are phosphorylated and remain in the cytoplasm, where they are inactivated. When the Hippo pathway is inactivated, non-phosphorylated YAP/TAZ enter the nucleus and act as transcriptional coactivators of other transcription factors [35]. YAP is reportedly *O*-GlcNAcylated under conditions of high intracellular glucose [11], which promotes YAP accumulation in the nucleus [36], consistent with our finding that nuclear localization is associated with poor prognosis. PFS is defined as the time until recurrence of malignant tumors and serves as an indicator of the effectiveness of treatment, and our data suggest that TAZ may be a useful new biomarker and therapeutic target in endometrial cancer. Tsujiura et al. investigated YAP expression and clinical data from 150 endometrial cancer patient specimens and identified increased nuclear YAP expression as an independent factor associated with poor prognosis [7]. Studies using two cell lines have found that both TAZ and YAP induce GLUT1 expression, suggesting that both YAP and TAZ may be indicators of poor prognosis, although further research is needed to address this possibility.

Recent studies have reported that, in other malignancies harboring KRAS mutations, the expression of YAP and GLUT1 is elevated [33]. Furthermore, it has been suggested that the Hippo pathway is involved in the development of resistance to KRAS inhibitors [37], and that combination with a TEAD inhibitor enhances the antitumor effect [38]. KRAS mutations have also been reported in endometrial cancer [39], suggesting that the results of this study may closely interact with KRAS signaling. Accordingly, therapeutic targeting of the Hippo pathway may represent a novel treatment strategy for endometrial cancer.

Although the findings of this study are promising, we recognize some limitations. In this study, only two Type 1 endometrial cancer cell lines (HHUA and Ishikawa) that are estrogen receptor-positive and exhibit an endometrioid histological type were used. Therefore, the results of this study may not be fully generalizable to Type 2 endometrial cancers, such as serous carcinomas or clear cell carcinomas. Further studies using a broader range of cell lines, including Type 2 models, are necessary to verify the universality of the observed mechanisms. The conclusion of this study is that using cross-sectional IHC data alone is insufficient to determine whether nuclear localization of TAZ is a factor that determines tumor malignancy or merely a marker of tumor progression. The in vitro data suggest that TAZ plays a functional role in gene regulation; however, further in vivo studies and longitudinal studies are necessary to clarify the causal relationship between nuclear TAZ and tumor progression. Ours was a single-center, retrospective study; therefore, further prospective, multicenter studies are needed to determine the clinical relevance of YAP/TAZ. We hope that large-scale cohort studies will verify that YAP/TAZ are predictors of poor prognosis in endometrial cancer and that this finding will lead to the development and clinical trials of YAP/TAZ inhibitors.

## 5. Conclusions

Our study demonstrated that YAP/TAZ control GLUT1 expression in two cell lines derived from endometrial cancer. Furthermore, our study confirmed a significant correlation between TAZ and GLUT1 expression in endometrial cancer patients and that TAZ expression is associated with poor prognosis. A graphical model of the proposed mechanism is shown in Figure 9, highlighting the role of Hippo pathway dysregulation in metabolic reprogramming and oncogenic signaling. These results suggest that YAP/TAZ promote tumor growth via GLUT1 and that therapeutic targeting of YAP/TAZ could therefore be useful in the development of future treatments.

## Figures and Tables

**Figure 1 cancers-17-02554-f001:**
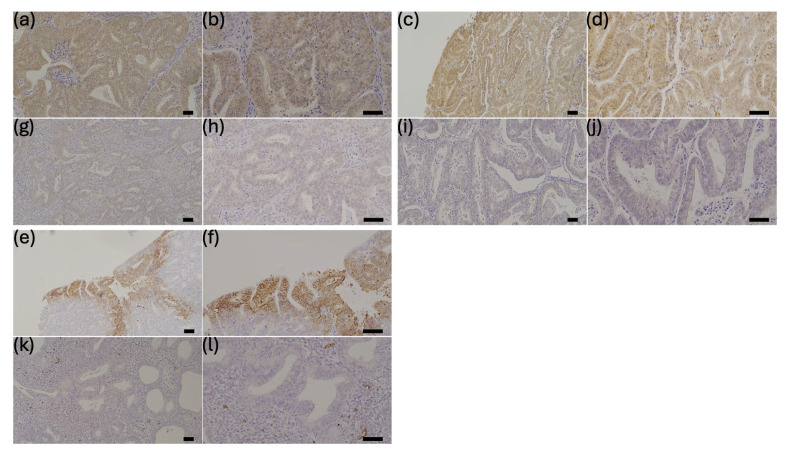
Representative sections of endometrial cancer tissue showing immunostaining for YAP (**a**,**b**,**g**,**h**), TAZ (**c**,**d**,**i**,**j**), and GLUT1 (**e**,**f**,**k**,**l**) (upper) and respective negative or weak staining (bottom) ((**a**,**c**,**e**,**g**,**i**,**j**): magnification ×100; (**b**,**d**,**f**,**h**,**j**,**l**): magnification ×200). Scale bar = 50 μm.

**Figure 2 cancers-17-02554-f002:**
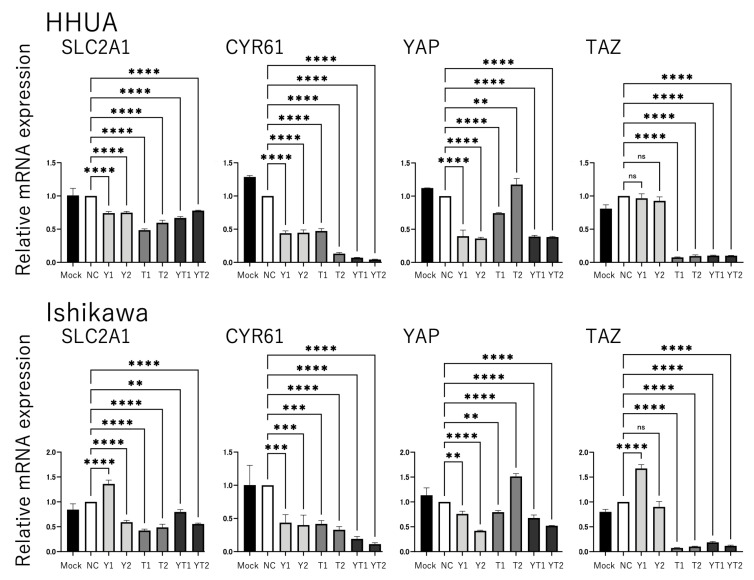
YAP and TAZ are regulators of GLUT1 (encoded by SLC2A1) expression in HHUA and Ishikawa cells. HHUA and Ishikawa cells were transfected with either of two independent siRNAs (YAP (Y1, Y2), TAZ (T1, T2), YAP/TAZ (YT1, YT2); 20 nM) or a non-targeting control (NC) (20 nM) siRNA. At 48 h after transfection, the respective effect of knockdown on the expression of SLC2A1, CYR61, YAP, and TAZ in HHUA and Ishikawa cells was analyzed. mRNA levels were measured by RT-qPCR and normalized relative to 36B4 mRNA. The ratio of the NC was arbitrarily defined as 1. Data are the mean ± SEM of three independent experiments. Statistical analysis was performed using the Sidak multiple comparison method (ns *p* > 0.05, ** *p* ≤ 0.01, *** *p* ≤ 0.001, **** *p* ≤ 0.0001).

**Figure 3 cancers-17-02554-f003:**
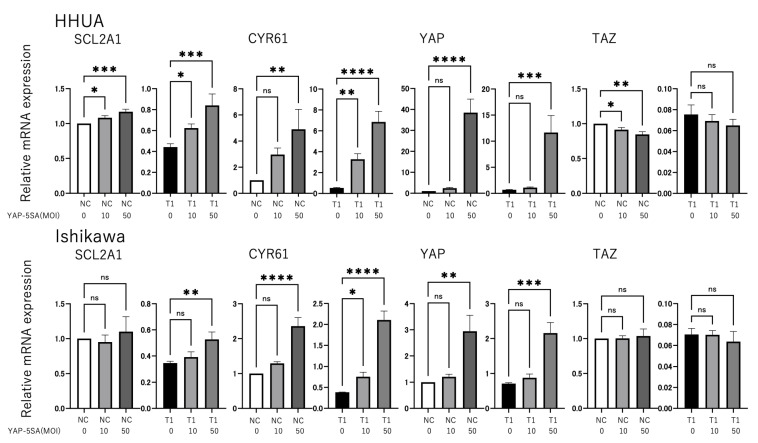
Introduction of constitutively active YAP induces GLUT1 (encoded by SLC2A1) expression in HHUA and Ishikawa cells. HHUA and Ishikawa cells were infected with adenovirus at the indicated multiplicity of infection (MOI), corresponding to the number of virus particles per cell. To combine siRNA knockdown and adenovirus-mediated transduction, adenovirus infection was performed 4 h after cells were transfected with TAZ (T1) siRNA (20 nM) or non-targeting control (NC) (20 nM) siRNA, and the cells were then cultured for an additional 48 h. The effect of YAP-5SA on the expression of SLC2A1, CYR61, YAP, and TAZ in HHUA and Ishikawa cells was then analyzed. mRNA levels were measured by RT-qPCR and normalized relative to 36B4 mRNA. The ratio of NC was arbitrarily defined as 1. Data are the mean ± SEM of three independent experiments. Statistical analysis was performed using the Sidak multiple comparison test (ns *p* > 0.05, * *p* ≤ 0.05, ** *p* ≤ 0.01, *** *p* ≤ 0.001, **** *p* ≤ 0.0001).

**Figure 4 cancers-17-02554-f004:**
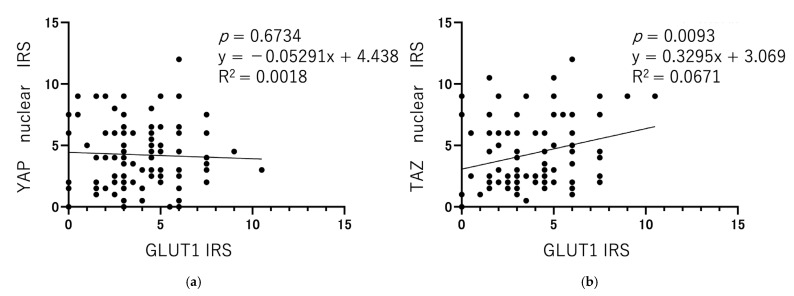
Correlation between YAP nuclear expression and GLUT1 (**a**) and TAZ nuclear expression and GLUT1 (**b**). Each point in the figure represents the IRS (n = 100) of each protein from clinical specimen tissues obtained from endometrial cancer patients (duplicates included).

**Figure 5 cancers-17-02554-f005:**
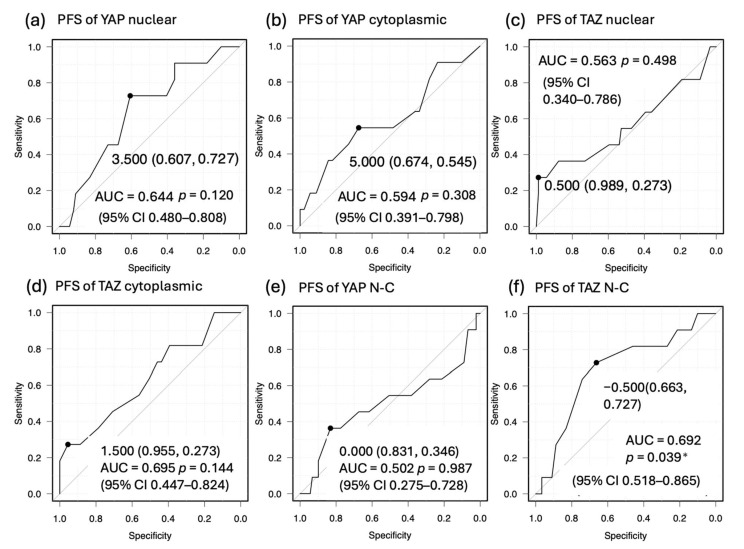
ROC curves for predicting PFS (**a**–**f**) based on IRS for nuclear YAP, cytoplasmic YAP, nuclear TAZ, cytoplasmic TAZ, YAP nuclear expression minus cytoplasmic expression (N-C), and TAZ N-C. The parentheses in the figure indicate (specificity, sensitivity). (* *p* < 0.05) (**a**) AUC = 0.644 (*p* = 0.120, 95% confidence interval (CI) 0.48–0.808), with an optimal cutoff value of 3.5 for nuclear YAP; (**b**) AUC = 0.594 (*p* = 0.308, 95% CI 0.391–0.798), with an optimal cutoff value of 5 for cytoplasmic YAP; (**c**) AUC = 0.563 (*p* = 0.498, 95% CI 0.340–0.786), with an optimal cutoff value of 0.5 for nuclear TAZ; (**d**) AUC = 0.635 (*p* = 0.144, 95% CI 0.447–0.824), with an optimal cutoff value of 1.5 for cytoplasmic TAZ; (**e**) AUC = 0.502 (*p* = 0.987, 95% CI 0.275–0.728), with an optimal cutoff value of 0 for YAP N-C; (**f**) AUC = 0.692 (*p* = 0.039, 95% CI 0.518–0.865), with an optimal cutoff value of −0.5 for TAZ N-C.

**Figure 6 cancers-17-02554-f006:**
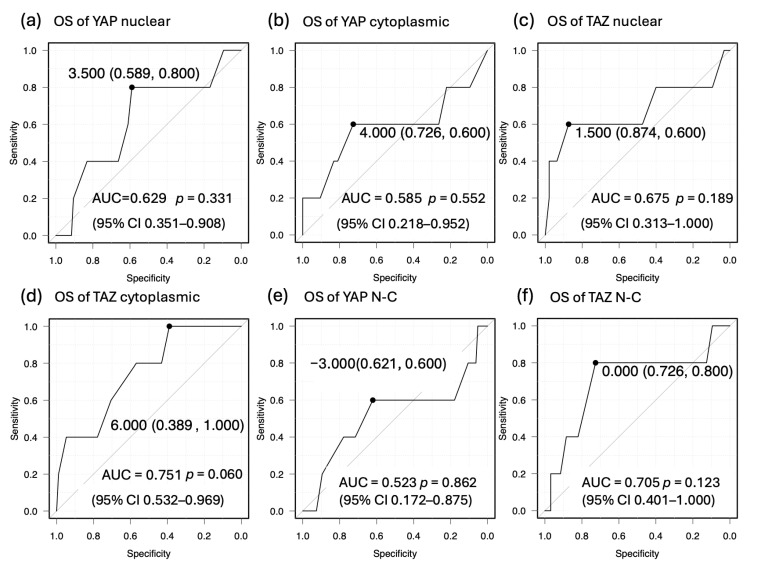
ROC curves for predicting OS (**a**–**f**) based on IRS for nuclear YAP, cytoplasmic YAP, nuclear TAZ, cytoplasmic TAZ, YAP nuclear expression minus cytoplasmic expression (N-C), and TAZ N-C. The parentheses in the figure indicate (specificity, sensitivity). (**a**) AUC = 0.629 (*p* = 0.331, 95% CI 0.351–0.908), with an optimal cutoff value of 3.5 for nuclear YAP; (**b**) AUC = 0.585 (*p* = 0.552, 95% CI 0.218–0.952), with an optimal cutoff value of 4 for cytoplasmic YAP; (**c**) AUC = 0.675 (*p* = 0.189, 95% CI 0.313–1.000), with an optimal cutoff value of 1.5 for nuclear TAZ; (**d**) AUC = 0.751 (*p* = 0.060, 95% CI 0.532–0.969), with an optimal cutoff value of 6 for cytoplasmic TAZ; (**e**) AUC = 0.523 (*p* = 0.862, 95% CI 0.172–0.875), with an optimal cutoff value of −3 for YAP N-C; (**f**) AUC = 0.705 (*p* = 0.123, 95% CI 0.401–1.00), with an optimal cutoff value of 0 for TAZ N-C.

**Figure 7 cancers-17-02554-f007:**
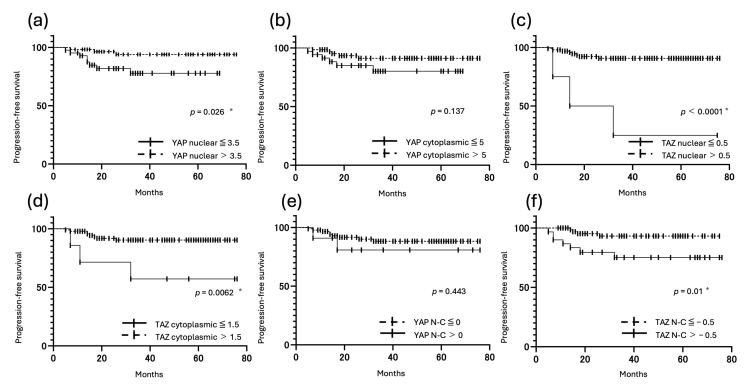
Kaplan–Meier survival curves for PFS (**a**–**f**) of patients with uterine carcinoma, according to the IRS for nuclear YAP, cytoplasmic YAP, nuclear TAZ, cytoplasmic TAZ, YAP N-C, and TAZ N-C. (* *p* < 0.05) (**a**) PFS of patients with low (≤3.5, solid line) and high (>3.5, dotted line) nuclear YAP IRS. (**b**) PFS of patients with low cytoplasmic YAP IRS (≤5, solid line) and high cytoplasmic YAP IRS (>5, dotted line). (**c**) PFS of patients with low TAZ nuclear (≤0.5, solid line) and high TAZ nuclear IRS (>0.5, dotted line). Patients with low TAZ nuclear IRS had poorer PFS than those with high TAZ nuclear IRS (*p* < 0.0001). (**d**) PFS of patients with low cytoplasmic TAZ IRS (≤1.5, solid line) and high cytoplasmic TAZ IRS (>1.5, dotted line). Patients with low cytoplasmic TAZ IRS had poorer PFS than those with high cytoplasmic TAZ IRS (*p* = 0.0062). (**e**) PFS of patients with high (≥0, solid line) and low (≤0, dotted line) YAP N-C IRS. (**f**) PFS of patients with high (≥−0.5, solid line) and low (≤−0.5, dotted line) TAZ N-C IRS. Patients with high TAZ N-C IRS had poorer PFS than those with low TAZ N-C IRS (*p* = 0.01).

**Figure 8 cancers-17-02554-f008:**
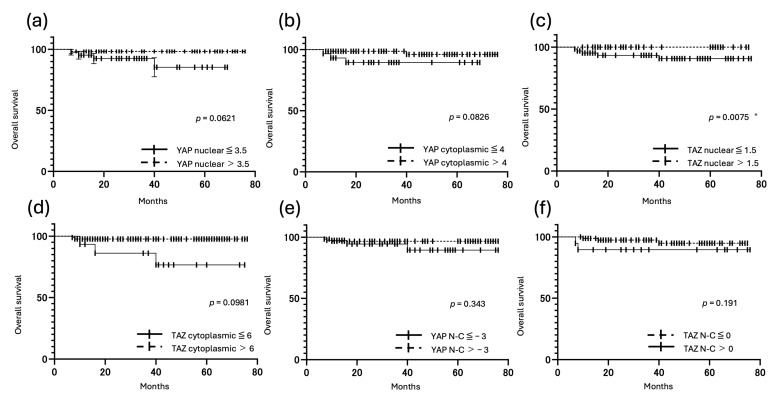
Kaplan–Meier survival curves for OS (**a**–**f**) of patients with uterine carcinoma, according to the IRS for nuclear YAP, cytoplasmic YAP, nuclear TAZ, cytoplasmic TAZ, YAP N-C, and TAZ N-C. (* *p* < 0.05) (**a**) OS of patients with low (≤3.5, solid line) and high (≥3.5, dotted line) nuclear YAP IRS. (**b**) OS of patients with low (≤4, solid line) and high (≥4, dotted line) cytoplasmic YAP IRS. (**c**) OS of patients with low nuclear TAZ IRS (≤1.5, solid line) and high nuclear TAZ IRS (>1.5, dotted line). Patients with low nuclear TAZ IRS had poorer OS than those with high nuclear TAZ IRS (*p* = 0.0075). (**d**) OS of patients with low (≤6, solid line) and high (>6, dotted line) cytoplasmic TAZ IRS. (**e**) OS of patients with low (≤−3, solid line) and high (>−3, dotted line) YAP N-C IRS. (**f**) OS of patients with high (>0, solid line) and low (≤0, dotted line) TAZ N-C IRS.

**Figure 9 cancers-17-02554-f009:**
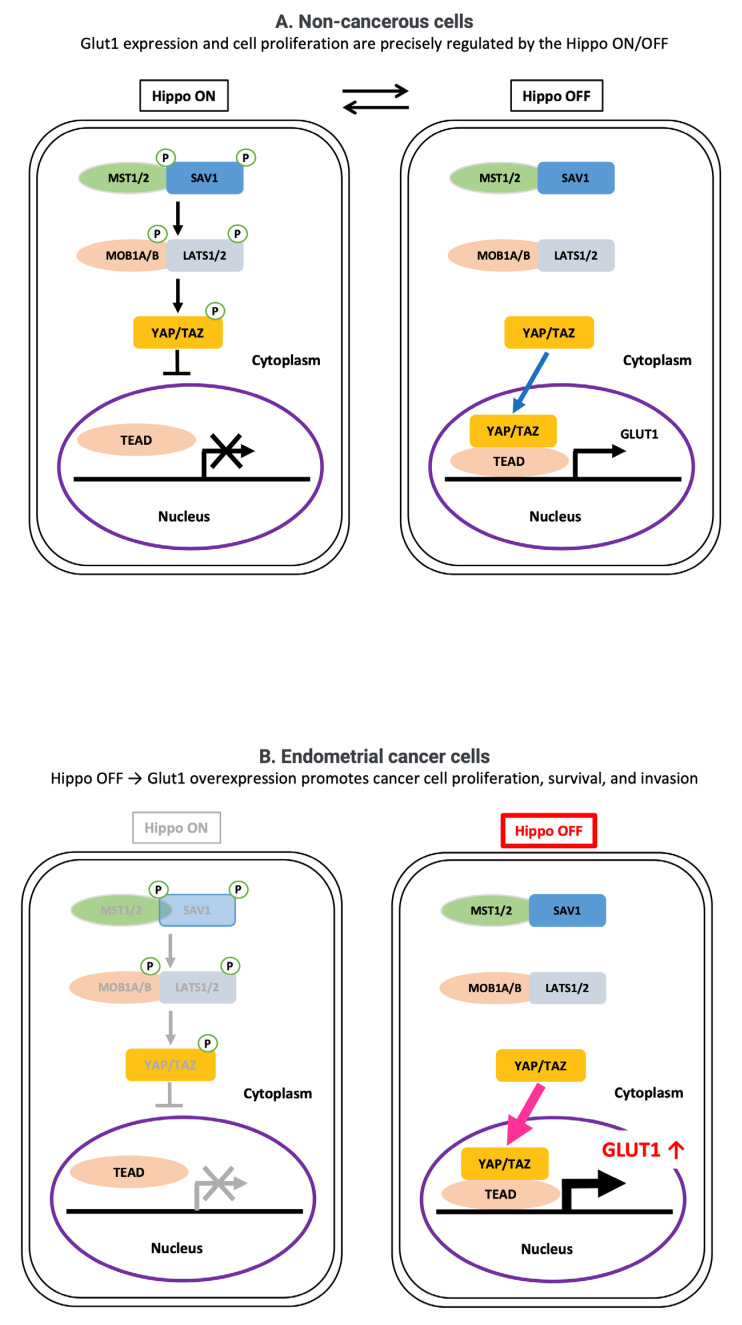
A schematic model of Hippo pathway regulation in non-cancerous and endometrioid cancer cells. Upper panel: In non-cancerous cells, the Hippo pathway regulates the dynamic subcellular localization of YAP/TAZ. When the Hippo pathway is ON, the MST1/2–SAV1 complex activates the LATS1/2–MOB1A/B complex through a kinase cascade, resulting in phosphorylation of YAP/TAZ. Phosphorylated YAP/TAZ (indicated by P in the figure) are retained in the cytoplasm and degraded, preventing their nuclear accumulation and downstream gene transcription. In contrast, when the Hippo pathway is OFF, YAP/TAZ remain unphosphorylated, accumulate in the nucleus, bind to TEAD transcription factors, and activate gene expression. Therefore, in non-cancerous cells, cell proliferation is controlled by turning the Hippo pathway ON and OFF. Bottom panel: Based on the findings of this study, endometrioid cancer cells exhibit reduced Hippo pathway activity (Hippo OFF), leading to the nuclear accumulation of YAP/TAZ and the enhanced transcription of target genes such as GLUT1. Upregulation of GLUT1 promotes glucose uptake and contributes to cancer cell proliferation, survival, and invasion.

**Table 1 cancers-17-02554-t001:** Patient and tumor characteristics.

Patient and Tumor Characteristics	
**Characteristic**	**n**
Total number of patients	100
**Histology**	
Endometrioid	
G1	65
G2	12
G3	13
Nonendometrioid	
Mixed	1
Serous	4
Carcinosarcoma	1
Clear	2
Mesonephric	1
Small cell	1
**Stage (FIGO 2008)**	
IA	58
IB	19
II	8
III	9
IV	6
**Treatment**	
Surgery	61
Surgery + chemotherapy	38
Lymphadenectomy	50
Hormone therapy	1
**Pathological features**	
Myometrial invasion ≥ 1/2	37
Tumor size ≥ 2 cm	75
Presence of LVSI	29
Presence of lymph-node metastasis	10
**Patient outcomes**	
Tumor progression	11
Death	5

**Table 2 cancers-17-02554-t002:** Relationship between nuclear YAP, cytoplasmic YAP, YAP N-C IRSs in tumors and clinical factors in endometrial cancer patients (* *p* < 0.05).

Variable	Patients (n)	Nuclear YAP		Cytoplasmic YAP		YAP N-C	
		Mean ± SE	*p*	Mean ± SE	*p*	Mean ± SE	*p*
**Age (y)**							
<50	12	4.792 ± 2.518	0.4756	6.667 ± 2.716	0.8315	−1.875 ± 2.001	0.6875
≥50	88	4.165 ± 2.615		6.5 ± 3.071		−2.335 ± 2.363	
**FIGO**							
I–II	85	4.418 ± 2.652	0.1156	6.818 ± 2.896	0.0197 *	−2.4 ± 2.22	0.1638
III–IV	15	3.233 ± 2.069		4.833 ± 3.244		−1.6 ± 2.804	
**Histology**							
Endometrioid G1 and G2	77	4.558 ± 2.565	0.0201 *	6.734 ± 2.84	0.1963	−2.175 ± 2.38	0.3211
G3 and others	23	3.174 ± 2.475		5.804 ± 3.528		−2.63 ± 2.112	
**Myometrial invasion**							
<1/2	63	4.595 ± 2.707	0.0934	7.095 ± 2.95	0.0179 *	−2.5 ± 2.584	0.2953
≥1/2	37	3.635 ± 2.314		5.541 ± 2.916		−1.936 ± 1.777	
**Tumor size**							
<2 cm	25	4.96 ± 3.119	0.1801	7.68 ± 2.802	0.0238 *	−2.72 ± 2.517	0.075
≥2 cm	75	4 ± 2.377		6.133 ± 3.006		−2.133 ± 2.247	
**LVSI**							
Absent	71	4.57 ± 2.631	0.0448 *	6.859 ± 3.026	0.0754	−2.289 ± 2.42	0.8157
Present	29	3.431 ± 2.371		5.69 ± 2.883		−2.259 ± 2.09	
**Lymph-node metastasis**							
Absent	90	4.406 ± 2.593	0.0479 *	6.722 ± 2.883	0.0493 *	−2.317 ± 2.255	0.5001
Present	10	2.75 ± 2.252		4.7 ± 1.181		−1.95 ± 2.948	

**Table 3 cancers-17-02554-t003:** Relationship between nuclear TAZ, cytoplasmic TAZ, TAZ N-C, and GLUT1 IRSs in tumors and clinical factors in endometrial cancer patients (* *p* < 0.05).

		Nuclear TAZ		Cytoplasmic TAZ		TAZ N-C		GLUT1	
Variable	Patients (n)	Mean ± SE	*p*	Mean ± SE	*p*	Mean ± SE	*p*	Mean ± SE	*p*
**Age (y)**									
<50	12	3.125 ± 1.932	0.0825	3.708 ± 1.751	0.009 *	−0.5833 ± 2.224	0.43	2.583 ± 1.222	0.0248 *
≥50	88	4.483 ± 2.76		5.705 ± 2.842		−1.222 ± 2.125		3.96 ± 2.172	
**FIGO**									
I–II	85	4.371 ± 2.629	0.6154	5.629 ± 2.798	0.2738	−1.259 ± 2.162	0.0537	3.665 ± 1.983	0.1648
III–IV	15	4.033 ± 3.176		4.533 ± 2.748		−0.5 ± 1.918		4.533 ± 2.768	
**Histology**									
Endometrioid G1 and G2	77	4.182 ± 2.6	0.428	5.461 ± 2.821	0.8432	−1.279 ± 2.21	0.0951	3.682 ± 2.042	0.3716
G3 and others	23	4.783 ± 3.037		5.478 ± 2.81		−0.696 ± 1.839		4.174 ± 2.391	
**Myometrial invasion**									
<1/2	63	4.068 ± 2.614	0.5881	5.754 ± 2.922	0.3203	−1.286 ± 2.28	0.0833	3.492 ± 1.883	0.1123
≥1/2	37	4.468 ± 2.763		4.973 ± 2.555		−0.905 ± 1.87		4.311 ± 2.425	
**Tumor size**									
<2 cm	25	4.44 ± 3.289	0.8658	4.94 ± 2.682	0.2579	−0.5 ± 2.638	0.438	3.12 ± 2.078	0.094
≥2 cm	75	4.28 ± 2.502		5.64 ± 2.84		−1.36 ± 1.913		4.02 ± 2.106	
**LVSI**									
Absent	71	4.197 ± 2.671	0.3974	5.345 ± 2.767	0.548	−1.148 ± 2.164	0.6691	3.507 ± 2.019	0.0421 *
Present	29	4.621 ± 2.805		5.759 ± 2.92		−1.138 ± 2.104		4.5 ± 2.248	
**Lymph-node metastasis**									
Absent	90	4.367 ± 2.651	0.6182	5.606 ± 2.767	0.2223	−1.239 ± 2.174	0.0218 *	3.8 ± 2.109	0.7996
Present	10	3.9 ± 3.264		4.2 ± 2.974		−0.3 ± 1.602		3.75 ± 2.383	

**Table 4 cancers-17-02554-t004:** Prognostic factors affecting progression-free survival in endometrial cancer selected by Cox univariate and multivariate analyses (* *p* < 0.05).

Variable	Univariate Analysis	*p*	Multivariate Analysis	*p*	Hazard Ratio (95% CI)	*p*
	Hazard Ratio (95% CI)		Hazard Ratio (95% CI)			
Age (≥50)	1.14 × 10^−11^	>0.9999				
FIGO stage (III–IV)	4.804 (11.36–302.7)	<0.0001 *	20.5 (4.29–152.9)	0.0006 *	21.63 (4.192–167.6)	0.0007 *
Histopathologic type (G3 and others)	20.91 (5.347–137.7)	0.0001 *	5.691 (1.169–42.67)	0.0484 *	6.738 (1.328–51.35)	0.0334 *
Myometrial invasion (≥1/2)	8.958 (2.306–58.79)	0.0051 *				
Tumor size (≥2 cm)	9.47 × 10^−5^	>0.9999				
LVSI (present)	7.499 (2.15–34.05)	0.0031 *				
Lymph-node metastasis (present)	31.01 (8.8–126)	<0.0001 *				
YAP nuclear (≤3.5)	3.97 (1.147–18.13)	0.0418 *	1.793 (0.4546–9.031)	0.43		
YAP cytoplasmic (≤5)	2.142 (0.6172–7.116)	0.2084				
YAP N-C (>0)	1.808 (0.2756–7.019)	0.1504				
TAZ nuclear (≤0.5)	3.296 (0.8614–10.95)	0.0576				
TAZ cytoplasmic (≤1.5)	0.3833 (0.05831–1.494)	0.221				
TAZ N-C (>−0.5)	4.338 (1.309–16.57)	0.0193 *			0.8589 (0.2123–3.862)	0.8335

**Table 5 cancers-17-02554-t005:** Prognostic factors affecting overall survival in endometrial cancer selected by Cox univariate and multivariate analyses (* *p* < 0.05).

Variable	Univariate Analysis	*p*	Multivariate Analysis	*p*	Hazard Ratio (95% CI)	*p*	Hazard Ratio (95% CI)	*p*
	Hazard Ratio (95% CI)		Hazard Ratio (95% CI)					
Age (≥50)	88,465,320,564	>0.9999						
FIGO stage (III–IV)	37.09 (5.164–749.2)	0.0016 *	26.6 (2.361–606.2)	0.0099 *	22.77 (2.106–503.1)	0.0122 *	36.88 (4.907–767)	0.0021 *
Histopathologic type (G3 and others)	3.59966 × 10^12^	>0.9999						
Myometrial invasion (≥1/2)	1.53315 × 10^12^	>0.9999						
Tumor size (≥2 cm)	33,601,241,091	>0.9999						
LVSI (present)	3.982 (0.6506–29.83)	0.1341						
Lymph-node metastasis (present)	10.57 (1.255–89.1)	0.0192 *	3.76 × 10^−5^	>0.9999				
YAP nuclear (≤3.5)	0.164 (0.008348–1.12)	0.1074						
YAP cytoplasmic (≤4)	0.2285 (0.2295–1.407)	0.1106						
YAP N-C (>−3)	0.4315 (0.05682–2.605)	0.3573						
TAZ nuclear (≤1.5)	0.127 (0.00167–0.7687)	0.0024 *	0.2493 (0.001093–2.843)	0.2758	0.1964 (0.00864–2.165)	0.1936		
TAZ cytoplasmic (≤6)	2.43 × 10^−12^	>0.9999						
TAZ N-C (>0)	3.099 (0.4080–18.71)	0.2155					2.486 (0.3137–16.37)	0.3361

## Data Availability

The data presented in this study are available upon reasonable request from the corresponding author.

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
