# Peer review of "YAP/TAZ Promote GLUT1 Expression and Are Associated with Prognosis in Endometrial Cancer"

_cancers, 2025, doi:10.3390/cancers17152554_

Round 1

Reviewer 1 Report

Comments and Suggestions for Authors

Dr. Yoshida and colleagues present a compelling manuscript detailing the role of YAP/TAZ signaling in promoting GLUT1 expression, along with a mechanistic exploration of how these factors correlate with prognosis in uterine endometrial cancer. The study is well-structured, and the hypothesis is thoughtfully developed throughout. It holds significant translational potential with meaningful therapeutic implications. However, several points need to be addressed before acceptance:

  1. Contextual Relevance: Recent studies have shown that YAP/TAZ signaling contributes to resistance in patients treated with KRAS inhibitors (PMID: 37729426), and GLUT1 has also been implicated in KRAS signaling pathways (PMID: 19661383). The authors should briefly discuss these findings to place their work within the current scientific landscape and enhance its relevance.

  2. Graphical Model: The authors should include a schematic model summarizing the key findings of their study, integrating known elements from the literature to illustrate how their results fit into the broader field.

  3. Figure Enhancement: In Figure 1, please include a zoomed-in inset to make the immunostaining signals more clearly visible and improve overall data presentation.

Reviewer 2 Report

Comments and Suggestions for Authors

The manuscript entitled “YAP/TAZ promote GLUT1 expression, and these factors correlate with prognosis in uterine endometrial cancer” investigates the role of Hippo pathway effectors YAP and TAZ in regulating GLUT1 expression and their prognostic significance in endometrial cancer. The study concludes that YAP/TAZ-driven GLUT1 expression may promote tumor growth, highlighting these molecules as potential therapeutic targets in endometrial cancer. Overall, this manuscript is well-structured and thorough, with high-quality figures. Thus, I recommend accepting the paper after the authors successfully address the following comments.

1. Title
The current title, “YAP/TAZ promote GLUT1 expression, and these factors correlate with prognosis in uterine endometrial cancer,” reads like a full sentence. Please revise it to follow standard scientific titling conventions and more concisely reflect the core focus of the manuscript.

2. Introduction
Please include a brief overview of the Hippo signaling pathway, outlining its main components, and how its dysregulation contributes to cancer progression, particularly in endometrial cancer.

3. Methods
In the “Cell Culture” section, specify the complete maintenance conditions for both cell lines used (e.g., temperature, CO₂ concentration) to ensure reproducibility.

4. Results
1) In Section 3.1, the data from Figures 2 and 3 should be described more thoroughly in the text to help readers interpret the findings more easily.
2) In Table 1, bold the category labels to enhance readability and visual clarity.
3) Please revise Table 2 to improve clarity.
4) In Section 3.4, expand the descriptions of Figures 5 and 6. When discussing AUC values, clarify what thresholds are considered indicative of good vs. poor predictive performance.

5. Discussion
Include a section discussing future research directions, including potential experimental strategies to validate YAP/TAZ as predictive biomarkers or therapeutic targets in endometrial cancer. Consider also addressing the translational potential of targeting the Hippo pathway or GLUT1 in therapeutic development.

6. References
Several references cited in the Introduction are over a decade old. Please consider updating the literature review to include more recent studies that reflect the current understanding of uterine endometrial cancer biology and Hippo pathway research, especially for the “Introduction” section.

Reviewer 3 Report

Comments and Suggestions for Authors

In the present paper, the authors report the “YAP/TAZ promotes GLUT1 expression, and these factors correlate with prognosis in uterine endometrial cancer. I would recommend the acceptance of this manuscript after minor revisions. Here are the following comments;

  1. How was the risk of off-target effects or compensatory activation of other pathways regulating GLUT1 expression addressed, and what steps were taken to guarantee the specificity of the siRNA-mediated reduction of YAP and TAZ?

  1. How can the alterations in GLUT1 levels can be firmly linked to YAP/TAZ activity in the tumour environment, considering that GLUT1 is widely expressed and controlled by a number of factors?

  1. The study used only two endometrial cancer cell lines (HHUA and Ishikawa). Are these cell lines good models for all types of endometrial cancer? If not, maybe just mention this as a limitation.

  1. Justify how TAZ nuclear localization is linked to a bad prognosis when nuclear TAZ is also linked to GLUT1, which is involved in metabolic support rather than direct oncogenic signalling?

  1. Why did nuclear TAZ expression lead to a worse progression-free survival, while nuclear YAP expression led to a better or unclear prognosis, even though they have similar molecular functions?

  1. How do you differentiate whether nuclear TAZ localization is a cause or consequence of increased tumor aggressiveness, especially given the limitations of cross-sectional IHC data?

Round 2

Reviewer 1 Report

Comments and Suggestions for Authors

accepted

Reviewer 2 Report

Comments and Suggestions for Authors

The authors have fully addressed the reviewers' concerns with substantive and thoughtful revisions in the manuscript. 
Currently, there are no major issues remaining. 
The revised manuscript is suitable for publication.